# The EFPA Test-Review Model: When Good Intentions Meet a Methodological Thought Disorder

**DOI:** 10.3390/bs8010005

**Published:** 2017-12-31

**Authors:** Paul Barrett

**Affiliations:** Cognadev Ltd., 18B Balmoral Avenue, Hurlingham, Sandton 2196, South Africa; paul@cognadev.com; Tel.: +64-21-415625

**Keywords:** measurement, EFPA guidelines, assessments, psychometrics, standards

## Abstract

The European Federation of Psychologists’ Associations (EFPA) has issued sets of test standards and guidelines for psychometric test reviews without any attempt to address the critical content of many substantive publications by measurement experts such as Joel Michell. For example, he has argued that the psychometric test-theory which underpins classical and modern IRT psychometrics is “pathological”, with the entire profession of psychometricians suffering from a methodological thought disorder. With the advent of new kinds of assessment now being created by the “Next Generation” of psychologists which no longer conform to the item-based, statistical test theory generated last century, a new framework is set out for constructing evidence-bases suitable for these “Next Generation” of assessments, which avoids the illusory beliefs of equal-interval or quantitatively structured psychological attributes. Finally, with no systematic or *substantive* refutations of the logic, axioms, and evidence set out by Michell and others; it is concluded psychologists and their professional associations remain in denial. As with the eventual demise of a similar attempt to maintain the status quo of professional beliefs within forensic clinical psychology and psychiatry during the last century, those following certain EFPA guidelines might now find themselves required to justify their professional beliefs in legal rather than academic environments.

## 1. Introduction

The European Federation of Psychologists’ Associations (EFPA: http://www.efpa.eu/) have generated a set of test-user standards and guidelines for psychological test reviews; currently published as the Revised EFPA Review Model for the Description and Evaluation of Psychological and Educational Tests, approved by the EFPA General Assembly in July 2013 [1]. The aim of the standards and review guidelines was to establish a set of common criteria and competencies which can be applied across Europe and indeed any other countries who wish to agree to the unified standards-setting and competency frameworks.

From lines 1–4 of the Introduction to the Guidelines, we see:
“The main goal of the EFPA Test Review Model is to provide a description and a detailed and rigorous assessment of the psychological assessment tests, scales and questionnaires used in the fields of Work, Education, Health and other contexts. This information will be made available to test users and professionals in order to improve tests and testing and help them to make the right assessment decisions.”

These are indeed good intentions. However, Joel Michell [2] had previously set out the constituent axiomatic properties of quantitative measurement upon which many of the EFPA psychometric guidelines rely, noting that no empirical evidence existed to support a claim that any psychological attribute varied as a quantity. His statements concerning the pathology of psychometrics seem to negate many of the good intentions of the EFPA committee.
“By *methodological thought disorder*, I do not mean simply ignorance or error, for there is nothing intrinsically pathological about either of those states …Hence, the thinking of one who falsely believes he is Napoleon is adjudged pathological because the delusion was formed and persists in the face of objectively overwhelming contrary evidence. I take thought disorder to be the sustained failure to see things as they are under conditions where the relevant facts are evident. Hence, *methodological thought disorder* is the sustained failure to cognize relatively obvious methodological facts …I am interested, however, not so much in methodological ignorance and error amongst psychologists per se, as in the fact of systemic support for these states in circumstances where the facts are easily accessible. Behind psychological research exists an ideological support structure. By this I mean a discipline-wide, shared system of beliefs which, while it may not be universal, maintains both the dominant methodological practices and the content of the dominant methodological educational programmes. This ideological support structure is manifest in three ways: in the contents of textbooks; in the contents of methodology courses; and in the research programmes of psychologists. In the case of measurement in psychology this ideological support structure works to prevent psychologists from recognizing otherwise accessible methodological facts relevant to their research. This is not then a psychopathology of any individual psychologist. The pathology is in the social movement itself, i.e., within modern psychology.”[2] (p. 374, Section 5.1)

This article examines the applicability and future relevance of the EFPA test review guidelines. Some of these guidelines are based upon *beliefs* about psychological measurement rather than evidence-based empirical facts; beliefs which are unsustainable given the axioms defining the constituent properties of quantitative measurement. This state of affairs may now invite a specific kind of legal challenge, should a psychologist present psychometric test scores and associated statistical information to a court without informing the court that the information provided is reliant upon the beliefs being true as stated. It is unfortunate that those drawing up these guidelines seemed unaware of the constituent properties of quantitative measurement and the consequences for the classical and modern psychometric test theory methodologies they promoted as “best practice” to other psychologists. Prior to 2013, Joel Michell, Mike Maraun, and Günter Trendler between them had published 20 or more articles and books elaborating on the failure of psychologists to acknowledge straightforward facts and logic concerning the requirements for the measurement of quantities. In the case of Trendler, an explanation was submitted as to why quantitative measurement was forever impossible to ever achieve in psychology. His latest article explains that even simultaneous additive conjoint measurement is impossible to implement with psychological attributes.

Given this state of affairs, I propose some rather more straightforward test review “frameworks” which are suited equally to the older kinds of questionnaires for which the EFPA guidelines were created, and to the newer generation of assessments which are now a commercial reality. These latter kinds of assessment do not conform to any kind of psychometric test-theory, but can still be evaluated using more appropriate and direct methods for reliability estimation and “results” validation. The new frameworks no longer rely upon an assumption of psychological attributes varying as quantities or equal-interval variables, nor that samples of items are drawn from hypothetical item-universes, or indeed any form of statistical test theory. However, they do share some of the more justifiable and sensible EFPA requirements for a psychological assessment to be considered acceptable for practical use.

## 2. A Critical Evaluation of the EFPA Test Review Guidelines

### 2.1. The EFPA Guidelines Which Are Sensible and of Pragmatic Utility

Part 1: Description of the Instrument, Sections 2–6, make a great deal of sense:General descriptionClassificationMeasurement and scoringComputer-generated reportsSupply conditions and costs

Likewise, Sections 12 and 13 of Part 2: Evaluation of the Instrument:Quality of computer-generated reportsFinal evaluation

### 2.2. Those Guidelines Which Represent a Target for Legal Challenge

The problems occur in Part 2, specifically: Section 10: Reliability and Section 11: Validity. These sections require the reporting/use of psychometric and statistical indices such as Coefficient Alpha, Pearson correlations, Factor Analysis, Structural Equation Modeling, Multi-Trait Multi-Method designs, and IRT methodology. All of which assume the item responses/attribute of interest vary with equal intervals or as a quantity. That is, the item responses or attribute being assessed is assumed to vary additively, with equal intervals between magnitudes, and in most cases employing decimal fraction magnitudes as in mean scores and calculations of variances etc. In IRT and factor-analytic methodologies, simple sum-scores of items to form scale scores are replaced with a constructed “latent” variable whose variation is sometimes reported to three decimal-place precision.

It is only when you carefully examine the axioms defining quantity, and the properties they require of attribute variation (as Michell has done), that you realize this whole statistical enterprise that constitutes classical and modern psychometrics is not just mistaken, but in fact exposes its protagonists to substantive legal challenge in a court of law. For example, take IQ; to present a mean IQ score to a court in 2-decimal place precision requires empirical evidence that the attribute in question (intelligence), for which the IQ score is the numerical estimate, does indeed vary according to the numerical real-valued, continuous numbers used to represent the mean “magnitude” of intelligence for an individual. The fact that there is no evidence that any psychological attribute varies as a quantity, or with equal intervals between magnitudes, and that no-one can explain the observable “intelligence” difference in an individual possessing say a score of 69 or 71, or indeed a score of 110 and 114, is an admission to the court that the number presented is based upon a belief (an assumption) about the precision of IQ/Intelligence “measurement”, not fact.

This is not to deny that we can make pragmatically useful judgments about the meaningfulness of differing magnitudes of psychological attributes, but these judgments/interpretations are confined to statements about “orders” of magnitude whose precision is relatively coarse. Furthermore, given the “common-or-garden” [3] or “everyday language” definitions of psychological concepts/attributes [4], and the confusion arising from the assessment of same-named attributes which barely agree with one-another’s scores [5,6], it is clearly unwise to present a test score to a court as “definitive” of a magnitude of a specifically defined construct. This point was made clearly by Michell [7].

In addition, while many point to IRT models as psychology’s answer to physical measurement, Wood [8] demonstrated that Rasch IRT models fit random coin-toss data perfectly. Some years later, Michell [9] explained why, having shown that fitting IRT models simply produces ad-hoc “constructed” variables whose meaning (i.e., what exactly is being assessed) remains undefined except by a priori assertion or by post-hoc external criterion investigation. In addition, as Maraun [10] has explained, these constructed attribute scales are not “latent variables” at all, but merely computationally constructed, item-specific, attribute scales derived from item response patterns.

Many who support the EFPA guidelines (and the psychometric methodologies which they specify as “required”) will suggest that ordered-class data are simply equal-interval magnitudes “measured” poorly; approximations if you like of underlying continuous, real-valued, attribute magnitudes. Once again, Michell [11,12] has shown how flawed such an assumption is; it would not stand scrutiny under adversarial cross-examination. Why is this? Because it is an assumption for which no empirical evidence exists to support its veracity. Furthermore, as Michell [7] has shown, the “quantity” assumption requires the differing observable magnitudes for a variable reflect the variations of a common cause of those magnitudes. As he concludes the abstract to his article:
“It follows that for attributes investigated in science, there are three structural possibilities: (1) classificatory attributes (with heterogeneous differences between categories); (2) heterogeneous orders (with heterogeneous differences between degrees); and (3) quantitative attributes (with thoroughly homogeneous differences between magnitudes). Measurement is possible only with attributes of kind (3) and, as far as we know, psychological attributes are exclusively of kinds (1) or (2). However, contrary to the known facts, psychometricians, for their own special reasons insist that test scores provide measurements.”

Tryon [13] in a response to Ferguson’s [14] article on the negative public perceptions of psychology as a “real science”, argued that for psychology to attain greater credibility:
“We need more, not less, by way of modern causal mechanisms”. Pennington (2014) asked and answered the question of what is required to provide a scientific explanation in the following way: “What does it mean to explain something? Basically, it means that we identify the cause of that thing in terms of relevant mechanisms” [15] (p. 3, emphasis added). Psychologists claim to have mechanism information, but as Tryon [16] and Tryon, Hoffman, and McKay [17] explain, these claims are mainly false and illusory. For example, the very popular biopsychosocial model is just a list of relevant factors; it explains nothing more about psychology and behavior than a glass–metal–petroleum model would explain about how automobiles work. Listing variables, or ingredients, does not constitute explanation. We place variable names in boxes and draw arrows among the boxes, thereby imputing causality that is never explained. Drawing arrows does not constitute explanation. We say that squared correlations explain variance when they only account for variance. Accounting is not explaining. We use brain scans to identify brain lobes that are associated with psychological functions, but we cannot explain how those brain lobes do anything psychological any better than phrenologists could. Associations are not explanations. We identify mediators by correlational methods and discuss them as causal mechanisms. Correlation cannot establish causation. These “explanations” are “illusions of understanding”.[13] (p. 505)

A generative causal mechanism is what enables the technical specification of the rules for the measurement of an attribute. Without any agreed-upon definitions of the precise meaning of any psychological attribute (which may be impossible), and with no obvious understanding of the constituent properties of quantitative measurement by those setting standards for “tests” and reviews, we are left with sections of clerical guidelines which represent the shared *guild-beliefs* about measurement rather than evidence-based recommendations.

The fundamental issue at stake here is about how we seek to acquire evidence that a proposed assessment of a psychological attribute is reliable and valid. That does not require us to commit to a variety of statistical or test-theory assumptions as recommended by the EFPA guidelines [18], but that we use appropriate methodologies to acquire evidence which is not reliant for its validity upon untenable “quantity” assumptions being true as stated.

We may even ask whether an attribute’s assessment in one culture or ethnic group is possible in another. However, this is not achieved using the conventional Structural Equation Modeling invariance analysis methods [19,20]. These configural, metric, and scalar approaches are reliant upon item responses and attribute magnitudes varying as quantities (or at least the ad-hoc latent variables constructed from them so varying). As Barrett [21] showed empirically and subsequently argued, such cross-cultural work cannot be achieved by structural configural analyses alone, even when using more appropriate order-relation configural analysis/matching procedures.

Another problem that arises when considering the EFPA guidelines on reliability and validity is the reliance of many of the psychometric indices and statistical test-theory methodologies on the concept of a “true score”. As Borsboom and Mellenbergh [22] explain, the concept of a true score is simply a convenient statistical proposition enabling computation of internal consistency reliability, standard errors of measurement, and confidence intervals on test scores. There is no empirical evidence that anyone possesses a “true score” on any psychological attribute, partly because of the imprecision of definition of an attribute, and the associated imprecision of any form of measurement of such an attribute. However, we can of course identify ranges of scores for individuals over retest durations which enable pragmatically useful ordered class judgments of score-stability. To go further than that again requires making assumptions about quantitative variation of attributes which are currently impossible to test [23].

Then there is the matter of construct validity. Surely what appears in the EFPA guidelines is no longer tenable given the publication of several major critiques of the entire concept [24,25,26,27]? These articles dovetail with Maraun’s [3] earlier exposition of the flaw with the psychometric conceptions of construct validation. Very recently, Slaney and Garcia [28] have also exposed the flaws with common conceptualizations of construct validity theory, with a recent book by Slaney [29] exploring matters in more detail. The point here is that the EFPA guidelines concerning construct validation are now obsolete, and were shown to be obsolete as far back as 1998.

### 2.3. How Have We Reached This Sub-Optimal State of Affairs

#### 2.3.1. The Aspiration to Emulate Measurement within the Physical Sciences

That is, for socioeconomic, political, and academic-status reasons, psychologists have chosen to adopt their own individual/characteristic definitions of measurement in order to create the impression among the wider academic and scientific community that psychology can make measurement in the manner and rigor in which it is made within the physical sciences [30,31,32]. As far back as the 1980s, Postman [33] severely criticized this ideal of psychologists as “social physicists”. In a very real sense, this seemingly delusional aspiration has perhaps led in part to the estimated average replicability rate of key-effects that top-tier psychology journals report—between 36% and 39% (objective vs. subjective rate; Open Science Collaboration; reported in in Zenker and Witte [34]). When the measurement of psychological attribute magnitudes is no longer aligned with how psychological attributes might vary, one of the outcomes to be expected will be lack of replicability of effects which are contingent upon those attributes varying homomorphically with the real number or integer relational system.

#### 2.3.2. The Commercial Imperative

When psychologists and publishers of tests found there was a profitable market for assessment products, the motivation to create financial rewards overcame any concerns about measurement. Instead, psychometric test theory and its methods were heavily promoted as the benchmark of any credible assessment evidence-base, which empowered professional psychological societies to impose standards for the training of test users. Such training proved to be highly profitable for test publishers while enabling them to promote themselves as “standards-bearers” for technical sophistication and rigor. In fact, psychometric complexity/expertise became a point-of-difference value proposition for many competitive sales approaches within the Human Resource field. The cold reality is that there is little obvious outcome-differentiation between the deployment of a DiSC, MBTI, or Brand X assessment which fails all psychometric evidence-base hurdles when compared with a CEB or Hogan psychometric masterpiece. This is to be expected given that all we can confidently claim is that psychometric test scores represent rank-orders of magnitudes, not quantities. The accuracy/precision accorded to test scores is illusory, which explains the lack of any tangible real-world organizational consequences when using typology assessments compared to “equal-interval” score-based assessments. The sad reality is that test-theory psychometric methodology has been reduced to the status of a marketing ploy; brought out to impress and amaze in competitive sales presentations, but with no-one evaluating the validity of the claims because few understand or agree upon what is actually being presented as “evidence”, given the level of abstraction of the various indices. Which in part explains Briner’s and Rousseau’s [35] conclusion that I/O psychology is not yet an evidence-based practice. Making untestable assumption-laden claims about measurement does not constitute evidence; at best it constitutes a kind of prevailing belief, at worst a purposeful deception.

#### 2.3.3. The ”Nelson” Syndrome 

From Wikipedia [36]:
“The phrase to turn a blind eye is attributed to an incident in the life of Admiral Horatio Nelson. Nelson was blinded in one eye early in his Royal Navy career. During the Battle of Copenhagen in 1801 the cautious Admiral Sir Hyde Parker, in overall command of the British forces, sent a signal to Nelson’s forces ordering them to discontinue the action. Naval orders were transmitted via a system of signal flags at that time. When this order was brought to the more aggressive Nelson’s attention, he lifted his telescope up to his blind eye, saying, “I really do not see the signal,” and most of his forces continued to press home the attack.”

Usually paraphrased as “I see no ships”, this quotation sums up the studious ignoring of the basic facts, logic, and axioms which define the measurement of a quantity. Those who formed the EFPA and the UK’s BPS and US’s APA committees chose to ignore clear facts concerning the constituent properties of quantitative measurement, and the consequences such facts might have on any guidelines they chose to issue. Instead they issued their guidelines and recommendations to other psychologists and organizations without any caveats.

This denial of reality serves no purpose other than to maintain a flawed status-quo which is increasingly unlikely to survive appropriate adversarial challenge in a court of law. There is now too much published counter-information on these issues, with no substantive published rejoinders. That intellectual silence speaks volumes, especially when a psychologist presenting test scores to a court is now required to justify the quantitative assumption upon which they rely for the validity of any test scores and quantitative statistical procedures offered to the court. The response “*I’m following best practice guidelines*” no longer suffices as a judge may realize that equating the measurement of a psychological attribute with that of a physics SI base or derived-unit measure [37] is not just unwise but untenable. The numbers used to represent magnitudes do vary additively; but whether or not the attribute itself does so remains unknown. Psychologists manipulate the numbers under the assumption that they represent equal-interval magnitudes of a psychological attribute; but they never seek to test that assumption.

### 2.4. Does Any of This Really Matter?

It might be argued that any set of systematic methodological procedures, such as those based upon an assumption that an attribute varies as a quantity, that “true-scores” can be invoked, that IRT can produce linear scales for measuring the magnitude of psychological attributes, is better than any possible alternative?

The problem is threefold:

❶ **Definitional:** We cannot define any psychological attribute with sufficient precision such that psychologists *en masse* agree on that single, precise technical definition, which in turn permits generation of the rules for measuring magnitudes of the attribute. Hence scales of varying numbers of “items” are generated which accord with a researcher’s personal definition of a “construct”, which entails the production of hundreds of scales all apparently measuring the same-named attributes but which only rarely show score-magnitude equivalence. Psychologists wander between single scales of items from the BFI-10 2-item Extraversion scale, the EPQR 21-item scale, through to the NEO-PIR 48-item scale (with six 8-item facets). All scales are referred to as “Extraversion”, but which one is the definitive measure of the “construct”? What about the MBTI assessment of Extraversion, or even the DiSC assessment, or CEB-OPQ’s “Extraversion” scale? The incredibly simple fact is we do not have a precise definition of the construct, one that determines a standard for the measurement of extraversion. Construct Validity? We might as well ask for Basil Fawlty’s sliced hippopotamus in suitcase sauce [38]! You might smile at this but consider the EFPA guidelines and their mantra-like doctrines on construct validity set out as “must haves”. A far more reasoned position is required; one that accepts the reality of a more general and imprecise “construct” definition and non-quantitative measurement.

❷ **Causal:** We have no causal theory for any attribute which causally explains, mechanistically, how variations in that attribute occur. As noted above in that quote from the article by Tryon [13]. None of the attributes we assess as psychologists possesses any mechanistic causal theory which specifies how magnitudes vary on that attribute. Therefore, we observe differences in magnitudes without any explanation how they have occurred; rather, we speculate broad causal effects without any attempt to empirically test such speculations. So how can we expect to claim an attribute varies quantitatively without any empirical evidence as to what could be causal for such precise, additive-unit, variation? As Trafimow [39] (p. 849) notes:
“On the other hand, in much, perhaps even most, research in psychology, there are no practically relevant units anyway. For example, in social psychology, what is an attitude unit, a prejudice unit, or a self-affirmation unit? In clinical psychology, what is a depression, anxiety, or psychopathy unit? In education, what is a knowledge unit? Similar questions can be asked with respect to measurement in many areas.”

❸ **Representational:** Psychologists persist in equating the results of arithmetic operations on the real or integer number system as though this numerical relational system is actually equivalent with the empirical attribute relational system. What if it isn’t?

Back in 1998, I set up a simple simulation in response to William Fisher’s and Ben Wright’s enthusiastic statements at the time, claiming the Rasch model could construct quantitative measurement from “poor” observations of an underlying quantity [40].

Here I comprehensively extended an example briefly presented by Fisher [41]—where I “measured” objects of known precise length using a “bad ruler” where the marks indexed ordered classes rather than quantity, but where each mark was assigned an equal-interval additive-unit magnitude (as psychologists do with test scores). I presented 40 “objects” for measurement to my bad ruler which consists of 16 unequal divisions of length … the objects are actually cm units on a real ruler, whose lengths are expressed using my bad-ruler units.

Each measurement is in the form of a dichotomy—a 1 is assigned to a bad measure unit if my cm “measure” extends beyond this unit. Where my cm measure is smaller than the remaining bad-ruler units, I assign a 0 to these units. The “responses” were “jittered”, adding some error (for the Rasch is a probabilistic model); the model dataset consisted of 40 “persons” and 16-item response vectors.

The rather simple-minded test here is whether the Rasch model will recover the equal-interval cm scale from the “ordinal” measures made by me. The example details and results begin at slide/page #26 of the 1998 presentation.

The results indicated that the Rasch difficulty parameters were “mirroring” my bad ruler integer units. If we did not know that my 16 numerically equal-interval integer “units” were in reality unequally spaced, we would refer to them as “equal-interval”, and plot them accordingly, noting that the Rasch difficulty/location parameters are now also equally spaced, mirroring the integer “units”.

I conducted a second simulation, to confirm my suspicion that the Rasch model is NOT able to reconstruct the underlying true linear equal-interval measurement from poorly observed data. Here, I mapped my “bad-ruler” units onto log_e_ (cm). Then I mapped equal-interval integers onto my units (ignoring the underlying non-linearity) to make my measurement observations as before. The results were the same; the Rasch modeling produced an equal-interval scaling of the additive-metric integer numerals which were indexing non-linear true cm lengths. The real unit of measurement (cm) was never “discovered” or revealed by the model.

Following this presentation in November 1998, Ben Wright from the MESA group at Chicago re-analyzed my data—and concluded that there was insufficient stochasticity (random error) in my observations. In short, my data may have been artificially “too clean” for the Rasch model to work as designed (A student informed me via email that they had replicated my results using both Rasch and 2-parameter IRT on a much larger simulation sample with more error added, and unsurprisingly obtained a similar result.) That response always struck me as somewhat bizarre, as Michell [8] (p. 126) later pointed out:
“Now, if a person’s correct response to an item depended solely on ability, with no random ‘error’ component involved, one would only learn the ordinal fact that that person’s ability at least matches the difficulty level of the item. Item response modellers derive all quantitative information (as distinct from merely ordinal) from the distributional properties of the random ‘error’ component. If the model is true, the shape of the ‘error’ distribution reflects the quantitative structure of the attribute, but if the attribute is not quantitative, the supposed shape of ‘error’ only projects the image of a fictitious quantitivity. Here, as elsewhere, psychometricians derive what they want most (measures) from what they know least (the shape of ‘error’) by presuming to already know it.”

It was Ben Wright’s “you need more error in your data” response, coupled with Wood’s [8] coin-tossing simulation that convinced me the Rasch model was not useful as a means of constructing quantitative measurement of a psychological attribute. All it is doing is scaling item responses which need to be observed with sufficient error in order to satisfy the conditions of the model. If you can observe responses/make observations without error (or you do so with a tiny amount of error as in physics), the Rasch model will not fit your data; not because it has somehow failed or is a “bad model”, but because the conditions required for it to have a chance of fitting data (heightened error) are not present. Such IRT scaling models as the Rasch model may have pragmatic value, but it is unclear what scientific value they may possess, given they are blind to the semantic content of items, and as Michell [8] notes:
“Item response modellers derive all quantitative information (as distinct from merely ordinal) from the distributional properties of the random ‘error’ component.”

The more random error in a dataset, the more likely a model will fit that dataset.

Constructing measurement with particular properties such as those required by a quantity requires detailed empirical experimentation comprising initial phenomenon detection, theory construction, and more experimentation (in an abductive process as Haig [42] describes) until a body of observations begins to consistently display those characteristics associated with quantity variation. This is aligned with how David Sherry [43] sees things with respect to the history of measuring temperature, and Mari [44] from within an engineering perspective.

In conclusion, all of the above is of great significance when it comes to making recommendations about “best practice” test construction/evaluation. Relying upon outdated thinking and assumption-laden test theory and statistics is neither wise, nor is it justifiable any longer. The EFPA guidelines in sections 10 and 11 are not current best practice at all, but merely repeat-prescriptions from a time when psychologists knew no better. As we will see below, the suggested new framework reflects the same kind of sentiment as that expressed by Michael Raynor and Mumtaz Ahmed summarizing the final results of the Deloittes’ Persistence Project [45] (p. 108) where their sample of enduring organizational performance data consisted of over 22,000 companies that traded on a U.S. exchange between 1966 and 2008:
“The many and diverse choices that made certain companies great were consistent with just three seemingly elementary rules:
Better before cheaper-in other words, compete on differentiators other than price.Revenue before cost-that is, prioritize increasing revenue over reducing costs.There are no other rules-so change anything you must to follow Rules 1 and 2.The rules don’t dictate specific behaviors; nor are they even general strategies. They’re foundational concepts on which companies have built greatness over many years.”

## 3. The Characteristic Features of the Next Generation of Assessments

What are the “Next Generation” of assessments? Well, these are, as you might expect, very different in design, look, feel, scoring, and administration, from the generation of assessments that began their life back in the early 1900s. We are all familiar with the current self-report questionnaires, ability-tests and their “puzzle-format” items, assessment center tasks, simple “psychometric” situational judgement tests and their variants. Indeed, many of these assessments and design concepts were originally created during the last century, and their characteristics and features perfected accordingly in line with associated developments in test theory psychometrics (as with item-response theory and structural equation modeling) and computing/display/internet technology.

However, an OPQ is an OPQ is an OPQ, however much you tinker with the items, display format, and scoring. Likewise, a WAVE, NEO, HPI, MBTI, 16PF, or 15FQ. Same as the WAIS, Stanford-Binet, Raven, MAB, Wonderlic, or other GMA assessments, and the various forms of clinical self-report/diagnostic questionnaires. In their own ways, these represent the “best of class” assessments in their chosen domain. They have been developed and honed over the years into their final forms, short or otherwise. However, their evolution was and is ultimately limited by a design that was conventional many decades ago. It was fashionable then but limited by the requirements and design-rules of test-theory psychometrics and the available technologies for delivering/deploying assessments; from paper-and-pencil supervised administration, hand-scoring, and expert-psychologist report writing through to the present-day mode of delivery of an assessment on a mobile/tablet with completely autonomous scoring and report generation. When you look back at the evolution of assessments, you cannot but be impressed at the sheer ingenuity, drive, and entrepreneurial skills and abilities of those who created the assessments and the current market for them. In essence, these academic and commercial innovators have collectively taken a world market amounting to a few thousands of dollars a year to many billions.

However, the psychometric product development process and test-industry was assigned a tombstone in 1998, in that now famous quotation from Sternberg and Williams [46]:
“No technology of which we are aware—computers, telecommunications, televisions, and so on—has shown the kind of ideational stagnation that has characterized the testing industry. Why? Because in other industries, those who do not innovate do not survive. In the testing industry, the opposite appears to be the case. Like Rocky I, Rocky II, Rocky III, and so on, the testing industry provides minor cosmetic successive variants of the same product where only the numbers after the names substantially change. These variants survive because psychologists buy the tests and then loyally defend them (see preceding nine commentaries, this issue).The existing tests and use of tests have value, but they are not the best they can be. When a commentator says that it will never be possible to improve much on the current admissions policies of Yale and its direct competitors [47] (p. 572, this issue), that is analogous to what some said about the Model T automobile and the UNIVAC computer. Comments such as this one prove our point [48] better than we ever could.”

However, no profitable commercial organization selling “Rocky” assessments was ever going to respond to this challenge; the marketing efforts and continued production of “psychometric” tests continued unabated, with the added impetus of IRT computer adaptive methodologies for delivering assessments. Indeed, even the ITC and associated test guidelines/standards authorities simply went about their business as usual, reinforcing and preserving the status-quo for the major test publishers.

However, the same ingenuity shown by the generations of the mid-to late 20th century was also being shown by a new generation of assessment developers during the 1990s, who with the advent of new powerful computing and display technology, a growing disillusionment with the “same-old” self-report assessments, and the burgeoning results from the field of cognitive psychology, began designing assessments which assessed psychological attributes from performance on *dynamic* tasks rather than performance on items [49]. Cognadev’s Cognitive Process Profile (CPP) was perhaps the very first serious commercial instantiation of such an approach, although it began first as a physical card-apparatus task before evolving into its computer-administered base version (later to be followed by the Learning Orientation Index for younger generations). Within the neuropsychological/cognitive diagnostics domain, the initial computer-based “performance” tasks were also being trialed, which are now part and parcel of routine modern neuropsychological “performance” assessment. Likewise, within the military, the Plymouth group of psychologists responsible for the British Army Ability Battery (BARB: [50]), were also experimenting with dynamic performance assessment.

However, now, things are very, very different. A clutch of new organizations such as Pymetrics, Knack.it, Revelian, Arctic Shores, Journey, and even the US Army, are now creating games played on mobiles and tablets to assess candidates for job-roles. EI Design, Merck, JobVille, Bluewolf, Cognadev, and many other smaller organizations/consultancies are providing gamified e-learning assessments. More established corporates such as L’Oreal (Clichy, France), KPMG (Amstelveen, The Netherlands), Deloittes Leadership Academy (New York, NY, USA), have built their own custom game-based assessments for internal assessment, leadership and talent management. Then there are the test publishers such as Cut-e, and start-ups such as Cliquidity (London, UK) and Cohired.com (Hamilton, ON, Canada), and more established organizations such as StaffCV and Findly, who provide biodata screens and psychometric assessments to individuals free of charge, so that the individuals can build their own portfolio of biodata and psychological assessments and be matched autonomously by these systems against paying-client organization job-vacancies. Finally, we have organizations such as Cambridge Analytica and the Psychometrics Centre (UK) selling psychographics; the provision of personality psychological attribute magnitudes acquired through behavioral-linguistic analysis of social networking internet activity of an individual. Hogan Assessments, through their innovation division, Hogan-X, are now advertising free online personality assessment using a variety of novel assessments tasks and stimuli (*such as facial-feature analysis*).

Apart from those organizations who enable individuals to develop their own assessment portfolio by completing established self-report psychometric assessments, the others who create gamified assessments are now producing assessments which are no longer aligned with the “psychometric standards” promoted by the ITC, BPS, APA, Veritas, or indeed any such “standards-based” organizations. Why is this? Because by their very nature, in the design, construction, and scoring-via-rule-based-algorithms/machine-learning, these next-generation assessments are no longer compatible with test-evaluation processes and methodologies which were created during the 20th century for 20th century tests.

The characteristic features of these new kinds of assessments are:They are designed from the ground-up using evidence-bases drawn from a wide vista of psychology, neuroscience theory, and experiment-evidence.There are no self-report/ability-test-type questionnaire items or static text-rich/scenario-type items (such as video or text-based Situational Judgment Tests); just information acquired from *dynamic* game or task performance, behavior, and/or linguistic analysis of text-based internet activity.Assessments sometimes are invariably comprised of thousands of very specific observations, clustered by expert-system rule or empirically into broader categories.Scoring is via expert-system rules and theory-relevant algorithms and/or varieties of machine-learning optimized prediction models.There are no psychometric “scales” as such, just attribute “type”, ordered-class, or quantitative (e.g., time) magnitude assessment which might be constituted from many diverse but theoretically/experimentally related behavioral information sources.Scoring information for any commercial assessment constitutes the Intellectual Property (IP) of an assessment, and is not made public.Reliability of any assessment outcome, type, ordered-class, or quantitative magnitude is assessed using appropriate retest methodologies only.Structural “validation” psychometrics (factor analysis, SEM modelling etc.) cannot be effectively used due to the mix of assessment-variable properties and expert-system rule-based scoring procedures which produce the final attribute magnitudes, orders, and classes. Basically, concepts from psychometric test theory simply do not apply to the kinds of attribute constituents, classifications, or “score” magnitudes.

Section 10 (Reliability), and especially Section 11 (Validity) of the EFPA guidelines appear somewhat irrelevant when looking at the list of the defining features of Next Generation assessments. This is not surprising because the guidelines were created for the previous generation’s assessments, whose design and construction were based upon a particular kind of “true-score” test theory and questionnaire-item-based measurement model (IRT). Given the untenable quantity assumption that pervades these guidelines, it is surely suggestive that now is the time to return to more straightforward and robust methods of evaluating the reliability and validity of a psychological attribute assessment; methods that are concordant with two *distinct* goals for its design and deployment: the Scientific and the Pragmatic. These two goals are completely discrete from one another.

## 4. The Proposed Next-Generation Test Review Frameworks

### 4.1. The Scientific Framework

From this perspective, an assessment is designed as part of a scientific investigation into a phenomenon, where the primary goal is to understand the cause/s of variation in some attribute; where that attribute is *the* fundamental constituent of the phenomenon. For example, if I am to propose a measure of a psychological attribute such as “*grit*’, the initial scientific task is to empirically identify the phenomenon of interest i.e., *grit*. This is phenomenon detection [42], and is the result of an initial set of observations which are subsequently interpreted to be “indicators” of *grit*. From this initial set of observations and meaning-assignment, causal explanatory theory construction can now be undertaken along with the construction of a prototype measure of *grit*, according to the rules now set down by an investigator for the measurement of *grit*. The evaluation of the measure requires a technical definition of what constitutes *grit* and what is causal for variations in the proposed measure of the construct. Then, it is a matter of empirical experimentation to demonstrate that what an investigator has proposed as causal for magnitudes of *grit* results in the numerical or otherwise defined magnitudes indicated by the assessment.

❶ If the causal theory for variation in *grit* proposes that magnitudes of *grit* must vary with equal intervals between its magnitudes, then a strong theory-claim is that *grit* varies as a continuous variable for which no normative upper or lower range-limiting bounds exist. Its magnitudes are additively structured but those magnitudes are assumed to be continuously varying over some arbitrary range (no agreed-upon minimum or maximum). Therefore, for a proposed measure of *grit*, these magnitudes are represented using the numerical real-number system but with the proviso there is no standard unit of measurement; hence if ratios of *grit* are constructed, these ratios are relative to the particular scaling of *grit* employed by an investigator.

❷ If *grit* is hypothesized as varying additively but not continuously, then a measure of *grit* would be represented using an equal-interval integer magnitude scale. Ratios of grit are no longer possible to compute except within the framework of integer arithmetic.

❸ If we propose that the best we can achieve is the identification/assessment of “orders” of *grit*, by dropping the additive-metric assumption, then we represent orders of *grit* via ascending-order symbols, which may be numbers or other ordered text-sequences such as letters of the alphabet.

Clearly, the experimental task of demonstrating that the psychological attribute *grit* varies additively with continuous or integer equal-intervals is going to be extremely difficult. To do so requires a technical/explicit definition of the construct; as within physics, a precise specification of what it is and how it should be measured, and what is causal or could conceivably cause such precise additive-unit variations in its magnitudes. Then this attribute is somehow manipulated experimentally, confirming that the variations so produced can be accurately represented using the real number or integer numerical relational system. The same “manipulation” issue applies if testing a proposed order-relation representation.

Indeed, this “manipulation requirement” is the very issue that Trendler [23,51,52] has addressed in his articles, the impossibility of manipulating the cause of any psychological attribute in such a way as to confirm/disconfirm the homomorphic relation to a numerical or order-relation system used to represent magnitudes of an attribute. It was thought this might be achieved by using additive/simultaneous conjoint measurement but Trendler [23] has shown that this is after all, impossible.

Although some, such as Carpenter, Just, and Shell [53] have developed computational/algorithmic causal process models “explaining” how a person might solve a particular kind of problem (in their case, Ravens matrix problems), these approaches are not designed to create an assessment or measure of a psychological attribute per se, but rather, provide a causal model explaining the occurrence (and variations in occurrences) of a particular phenomenon. The recent computational causal model of how children learn and solve problems in fractional arithmetic [54] is another such model. The goal is not to design an assessment per se, but to design a computational causal model which explains variations in human performance.

However, some investigators do set out to create an assessment which is based upon a technically-specified causal model for variations in a psychological attribute. For example, assume we have created an assessment of information processing speed (scored 0–120 s) based upon an assessment involving the manipulation of text-information by an individual where time taken to correctly manipulate the information is “the measure”. From a scientific perspective, to evaluate the claim that the assessment is measuring “information processing speed’, we might employ an experiment task where we directly control stimulus information in terms of formal “bits” of information required to be processed to complete the task, and note the time taken to correctly complete it expressed as bits-per-second processing speed. A simple criterion task might be a visual-stimulus choice reaction time. However, we are already assuming that what is meant by “information processing” is the same for both tasks. What is missing is that information processing speed as a psychological attribute, measured in seconds, may not be related in any straightforward way to sensory information processing speed measured in *information-bits-per-second* processing. In addition, visual stimulus choice reaction time is a very specific form of sensory information processing. We might even be tempted to utilize Lehrl and Fischer’s [55,56] Basic Information Parameter (BIP) as our information-bits-per-second criterion of verbal processing speed, if this is proposed as being a fundamental measure of intelligence. However, this measure was subsequently found to be little more than reading speed [57], which is related to but not definitive of variations in human intelligence.

Basically, there are no specific procedures for evaluating the reliability or validity of an assessment that is constructed within a scientific framework. Rather, the very principles and aims of conducting scientific research in an area impose a set of constraints on evaluation that are self-obvious from the very nature and purpose of scientific enquiry.

### 4.2. The Pragmatic Framework

With a typical pragmatic assessment such as the actuarial estimation of the recidivist risk of an offender, the constituent “items” in a scale are “instances” or “exemplars” of behaviors and events considered indicative of some important outcome. The “magnitude” scale is formed from the sum of binary or pragmatically weighted responses to these items rather than assuming that an underlying “latent” variable is causal for variations in the responses. Likewise, employee safety risk, where various indicators of risk are combined algorithmically; the goal being to form an accurate probability estimate of an employee’s safety risk optimized against observed workplace incident data [58]. From this perspective, an assessment is designed to assess a pragmatically useful attribute or the likelihood of an outcome without any attempt to create an a priori model which explains the causes of the observed magnitude variations within individuals.

However, within the psychometric assessment domain, the current *modus-operandi* builds attributes from descriptive aggregate “structural” statistical analyses of item covariance data (factor analysis, cluster analysis, non-metric MDS etc.), via phenomenon detection, or indeed any synthesis of ideas/experiment observations/broad theoretical viewpoints which enable the creation of what is usually considered a “latent variable” attribute magnitude scale. In this assessment construction process, the rules for the constituent properties of measurement of the attribute are always left unstated; the generic assumption is that the magnitudes of any psychological attribute assessment must be “equal interval”—as was stated succinctly by Frank Schmidt in a comment on ResearchGate, dated 26 July 2017 [59].
“The argument about ordering vs. quantification has been made about all social science research. It does not hold water. There is plenty of evidence that many psychological scales are essentially interval scales or close to it”.

When requested to provide the empirical evidence substantiating such a claim, no response was forthcoming. A revised version of the article is now published which drops the phrase “Intellectual Honesty” [60].

In essence, this is the default view for the vast majority of psychological test constructors. It reflects the mindset of the previous generations of 20th century psychometricians, where classical and modern test theory psychometrics was predicated on the assumption that all psychological attribute magnitudes vary as continuous or integer equal-interval entities; the very basis in fact of all metric psychometric test-theory. Put simply, the view is one stop short of claiming that these attributes vary as SI-base and derived unit quantities. In the 21st century, we now know better; and importantly we know this assumption will not survive scrutiny in legal settings where evidence, not assertion, is required to support specific measurement claims.

Thus, we now find ourselves in what is essentially undiscovered country; pioneers if you like. We know that any assessment must be evaluated for its reliability but it must also possess validation evidence which justifies its use (and utility). We also know that the next-generation assessments which do not conform to clerical checklist guidelines are here with us right now. In addition, we also now know the old EFPA guidelines concerning reliability and validity assessment are no longer fit for purpose in assessing the modern clutch of instruments which are rapidly becoming the user-norm within the world of personal and organizational development.

How do we then pragmatically evaluate the next generation of assessments? Let me set out the framework—because that is all it is. A framework of requirements. There are no psychometric or formulaic calculations/steps to follow. Assessment evaluation is designed around how an assessment is constructed and delivered, what it reports, and the claims made for what it reports. I’m following Deloittes’ Persistence Project Triple Crown analogy [61] (p. 28) where the “Triple Crown” concept is drawn from US baseball and horse racing’:
“Baseball and horse racing provide two popular analogues. Baseball batters with the greatest number of home runs, runs batted in (RBIs) and highest batting average in a year are triple crown winners. In U.S. horse racing, the triple crown goes to the 3-year-old thoroughbred that wins the Kentucky Derby, the Preakness Stakes and the Belmont Stakes in the same year.”

And how the authors summarize their analyses of data revealing these “Crowns” for organizations seeking to succeed:
“In the end, triple crown companies share the following attributes:
Clarity of visionDisciplined resource allocationExcellence in execution” p. 39

No formula, no specific rules, no specific steps to follow—just three overriding principles shared by organizations possessing enduring rather than short-term success.

Therefore, with respect to assessment standards/evaluation guidelines, I’m proposing three key requirements that need to be addressed by any assessment developer, test publisher, or evaluator/assessor/user:

**R1**: **The Measurement Assumption**: As there is no evidence to suggest any psychological attribute varies as a quantity, or even equal-interval of any description, we should not make that assumption. The most reasonable assumption is that we can assess partial orders or classes with some degree of “fuzziness” between boundaries. Without any clear methodology for determining precisely how an attribute varies and what is causal for those variations, we are relying upon “common-sense” judgement of magnitude, allied to some evidence in **R3** below. By all means we may use numerical sum scores, algorithmically-defined magnitude scales, whatever is suitable and convenient, but always with the proviso that these are for sheer convenience rather than an indication of precision. It is important to avoid such assumption-laden work by substituting orders, class-based, and actuarial analyses wherever it makes good sense to do so.

**R2**: **Evaluate Reliability**: There is only one measure—repeatability/reproducibility. Otherwise known as retest. Alpha and Omega indices are no longer remotely relevant. Retest reliability requires an answer to the question: “*will an individual obtain the same score if retested today, tomorrow, next week, next month, next year, over whatever duration is considered relevant?*” Ordinarily, retest reliability and a discrepancy-score/class analysis would answer this question nicely, except that retest reliability estimation now includes three potential threats to the accurate estimation of a reliability parameter:Non-systematic random error associated with the internal integrity of the test itself.Systematic and meaningful attribute variation of the attribute over periods of time within and extending across the retest duration.Memory of previous responses artificially causing consistency in 2nd occasion response patterns.

The end result of a retest analysis would nevertheless be an indication of reliability, though the causes of any substantive unreliability are not able to be disentangled from each other except by further careful empirical investigation.

When working with numerical approaches to reliability estimation, magnitude trumps monotonicity. That is, Pearson or Pearson-based correlation assessments are only used as indicators of monotonic relationship. They are not estimates of reliability because magnitude information is lost as part of the calculation process. Reliability is about score-magnitude equivalence, not indexing an order-relation (monotonicity).

When working with orders or types, existing or new methods directly suited to demonstrating repeatability of order or type are used. The logic of such methods is always clear-cut to explain and justify because no attempt is ever made to extrapolate to hypothetical sampling distributions, item universes, populations of individuals, or true scores. If an assessment provides its results in terms of what type a person is, then that is sole focus of the reliability assessment, answering the question: “*will an individual obtain the same type-assignment if retested today, tomorrow, next week, next month, next year, over whatever duration is considered relevant?*” The evidence base for reliability is constructed via suitable initial investigations and a body of empirical replications of reliable results.

**R3**: **Validation**: No more or less than providing empirical evidence in support of any claim that is made regarding the results reported by an assessment. If an assessment assigns a type-classification for an individual, then evidence is required to justify the accuracy of that assignment, answering the question: “*what evidence is there that the type assigned by the test for an individual is an accurate assessment of that individual?*” External data are required which corroborate the type-description based upon the responses or behaviors within an assessment that have been used to assign a type. If the assessment provides an ordered-class “score” or magnitude, then “external-to-the-assessment” evidence is required to substantiate the claim that those ordered magnitudes are indicative of the meaning associated with the assigned attribute magnitudes. This translates to detailing evidence that assessment magnitudes are aligned with externally observed phenomena which are indicative and/or aligned meaningfully with the assessment magnitudes. For example, if an assessment reports ordered-magnitudes of what it refers to as *grit*, then what is required is evidence indicating that individuals possessing each level of magnitude can be differentiated from others by the qualities or frequencies of behaviors they exhibit which are deemed indicative of a particular magnitude of *grit*. If an assessment constitutes a game wherein game-playing/interactive behaviors are deemed indicative of particular characteristics of an individual, then *external-to-the-game* empirical evidence is required to demonstrate the validity of claims phrased as “*this measures/estimates/indicates X*”. Clearly, if an assessment provider claims that the attribute class, order, or numerical score is predictive of some outcome, then relevant empirical evidence quantifying that predictive accuracy has to be forthcoming in order to justify such a claim.

To justify a claim that an assessment constructed for use in a particular culture is applicable to another, then it is a matter of evaluating the evidence that supported the use of the assessment in the originating culture alongside the comparable evidence which is proposed as supporting its use in another culture. Given **R1**, we cannot assess “invariance” or other qualities that might be applicable to quantitative variable assessment. Thus, we have to build a “case” for our assessment. This case would demonstrate the validity of a claim that an assessment which was, for example, validated in the US, possesses similar response behaviors and the same or similar expected phenomenal outcomes in say Spain. Admittedly, this is a complex area because while an assessment may indicate the same magnitude of an attribute in two cultures, how that magnitude is reflected in say frequencies of relevant behaviors may be different, conditioned by particular cultural influences.

A pragmatic evaluation requirement is concerned with constructing an evidence-base for an assessment’s claims which would stand scrutiny under adversarial examination in a court of law. Because no evidence exists that any psychological attribute varies as a quantity, use cannot be made of methods of evidence-base construction which rely upon this untenable and untested assumption. Which is why a “case-building” approach to validation is essential. If on the basis of our assessment, we claim a person will be the type of person who will show “good judgement”, then clear empirical evidence is required to show that indeed such people can be differentiated by the defining phenomenon of interest (good judgements) from those assigned a lower magnitude or class of judgement. In short, as David Freedman [62] argued, we need to use our “shoe-leather” to construct empirical evidence-bases justifying our “measurement” claims.

For many reading the above, the terms: construct, content, face, predictive, concurrent, and ecological validity are missing from the validation process. At best, the validation seems to be all about concurrent/predictive validity. This is because these terms are now entirely obsolete and irrelevant, along with “nomological nets”. Borsboom, Mellenbergh, and van Heerden [63] (p. 1061) provided the definitive arguments and clarifications for what constitutes validity and validation:
“Validity is not complex, faceted, or dependent on nomological networks and social consequences of testing. It is a very basic concept and was correctly formulated, for instance, by Kelley [64] (p. 14) when he stated that a test is valid if it measures what it purports to measure …A test is valid for measuring an attribute if and only if (a) the attribute exists and (b) variations in the attribute causally produce variations in the outcomes of the measurement procedure.”

In addition, with regard to the distinction between *validity* and *validation*:
“This is clear because validity is a property, whereas validation is an activity. In particular, validation is the kind of activity researchers undertake to find out whether a test has the property of validity. Validity is a concept like truth: It represents an ideal or desirable situation. Validation is more like theory testing: the muddling around in the data to find out which way to go. Validity is about ontology; validation is about epistemology. The two should not be confused.”

Finally, construct validity as a concept was shown to be deeply flawed by Maraun [3] in 1998, and was finally rendered obsolete by the powerful arguments in two chapters authored by Borsboom and colleagues and Michell [24,25] in a 2009 book on test validity. All of which has been completely ignored by psychometricians and those teaching practitioners.

### 4.3. A Matter of Scientific Integrity

Understandably, many readers might view these “Next Generation” requirements as a hugely damaging retrograde step; the relinquishing of the quantitative precision, test theory, methodologies, and statistical models/rigor of several generations of statisticians and psychometricians. However, these were all predicated upon an idealized view of measurement in psychology that no longer possesses any serious credibility. As I noted in the Introduction, the challenge is that presented by Joel Michell [1] (p. 374):
“I am interested, however, not so much in methodological ignorance and error amongst psychologists per se, as in the fact of systemic support for these states in circumstances where the facts are easily accessible. Behind psychological research exists an ideological support structure. By this I mean a discipline-wide, shared system of beliefs which, while it may not be universal, maintains both the dominant methodological practices and the content of the dominant methodological educational programmes. This ideological support structure is manifest in three ways: in the contents of textbooks; in the contents of methodology courses; and in the research programmes of psychologists. In the case of measurement in psychology this ideological support structure works to prevent psychologists from recognizing otherwise accessible methodological facts relevant to their research. This is not then a psychopathology of any individual psychologist. The pathology is in the social movement itself, i.e., within modern psychology.”

For those in the International Test Commission, those who constructed the relevant EFPA guidelines in question here, and indeed any professional society which issued guidelines for evaluating assessments based upon that “equal-interval” assumption which pervades psychometrics, the key question to be answered is:
“Why did you proceed with guidelines post-1997 which included methods of assessment evaluation predicated upon an untenable and untested assumption?”

The end result is courts being presented with latent variable or average IQ scores expressed to two-decimal place precision, as was the case recently in the US Supreme court where the decision relating to the death penalty for an offender rested *in part* upon such illusory precision. The average IQ presented to the court was 70.66, with various “standard error and confidence intervals also being presented for each individual cognitive ability assessment. All these statistical and test-theory “adjustments” are themselves predicated upon IQ varying as an equal-interval attribute [65]. Given there is no empirical evidence IQ varies as a quantity, or indeed varies as an equal-interval attribute [7], case-law has developed in many countries based upon a belief about measurement promoted exclusively by many psychologists, instead of being based upon empirical evidence supporting the claims of measurement. The potential legal consequences of this state of affairs are profound.

However, it might be argued that the new methodology termed “computational psychometrics” is the evolution of test-theory psychometrics for the 21st century and newer Next-Generation assessments [66,67]. Look very closely at how Polyak, von Davier, and Peterschmidt [68] (p. 5) define computational psychometrics:
“Computational psychometrics (CP) is defined as a blend of data-driven computer science methods (machine learning and data mining, in particular), stochastic theory, and theory-driven psychometrics in order to measure latent abilities in real-time”.

The utilization of computer science/machine learning methods to form probabilities of “beliefs” (as in the Bayesian Belief Networks used by the authors) mimics an actuarial approach to forming probabilities of outcomes as is seen within the entirely non-metric Violence Risk Assessment Guide [69] and other actuarial models of offender adverse-outcome risk prediction. However, again in this *computational psychometrics*, we see latent variables being created whose magnitudes vary as real-valued continuous quantities, not necessarily because there is empirical evidence that they do so, but because the authors simply decide that they will do so. It may be others will forgo the illusory precision of creating such quantitative latent variables, as that assumption-laden move is entirely unnecessary.

Ultimately, the more innovative Next-Generation psychologists are adopting a new honesty about their actual capability of measuring any psychological attribute as a quantity or equal-interval-varying attribute. They follow in the footsteps of Richard Feynman’s [70,71] famous words, reprinted in James Wilson’s [72] Doctoral colloquium keynote address:
“In the South Seas there is a cargo cult of people. During the war they saw airplanes land with lots of good materials, and they want the same thing to happen now. So they’ve arranged to make things like runways, to put fires along the sides of the runways, to make a wooden hut for a man to sit in, with two wooden pieces on his head like headphones and bars of bamboo sticking out like antennas—he’s the controller -and they wait for the airplanes to land. They’re doing everything right. The form is perfect. It looks exactly the way it looked before. But it doesn’t work. No airplanes land. So I call these things cargo cult science, because they follow all the apparent precepts and forms of scientific investigation, but they’re missing something essential, because the planes don’t land.Now it behooves me, of course, to tell you what they’re missing… It is a kind of scientific integrity, a principle of scientific thought that corresponds to a kind of *utter honesty*—a kind of leaning over backwards. For example, if you’re doing an experiment, you should report everything that you think might make it invalid—not only what you think is right about it: other causes that could possibly explain your results; and things you thought of that you’ve eliminated by some other experiment, and how they worked—to make sure the other fellow can tell they have been eliminated… In summary, the idea is to try to give all of the information to help others to judge the value of your contribution; not just the information that leads to judgment in one particular direction or another.”[72] (p. 1405)

The new pragmatic “framework” set out above is my interpretation of what that kind of “utter honesty” entails. Yes, the illusory precision is lost, but what is gained is a new quality and substantive realism surrounding the use of evaluative/analytical methods that far exceeds the assumption-laden claims of current psychometricians. It is hard, pioneering work, focused on constructing realistic and legally-defensible evidence-bases, while avoiding those EFPA pillars of psychometric “guidelines” whose foundations now more resemble a house of cards.

## 5. Concluding Remarks

### 5.1. The Legal Challenge to EFPA Guidelines

It is perhaps prudent to consider the example from clinical forensic psychologists and psychiatrists, who for decades had maintained that the accuracy of their clinical judgement was not a matter for empirical investigation, and who rigorously maintained that position among themselves and their professional societies. They found themselves faced with a challenge to their beliefs, not by other academics or practitioners (who they could simply ignore or ridicule), but by the courts in which they routinely presented their clinical judgements as “evidence”. Jay Ziskin [73] released the first edition of what eventually became the famous three-volume handbook for adversarial counsel, in which he set out how to question psychologists and psychiatrists in such a way in court, so as to reveal the beliefs they were presenting to the court as “evidence” [74]. The chapter in this 5th edition on challenging clinical judgement is a masterpiece in terms of the sheer body of evidence used in adversarial questioning to undermine the metaphorical hand-waving of experts providing testimony to the court. By the late 1980s, early 1990s, the success counsel were enjoying worldwide in the dissembling of clinical judgement/beliefs presented as “evidence” to the court energized some forensic and clinical psychologists to develop “evidence-based/actuarial” tools which presented risk-estimates to the courts as empirical facts. Clinical judgement now had to be based upon factual evidence of risk, where the judge as the “trier of fact” could establish the veracity of the formulation of risk taking into account empirical evidence as well as the clinician’s formulation using such evidence. It is no exaggeration to conclude that Ziskin created a revolution in terms of how “experts” in future would present their judgments and formulations to a court of law.

### 5.2. The “Next Generation” Assessment Challenge 

With the advent of new assessment and interview technologies, new highly automated recruitment and selection strategies, and the use of Big Data and Machine-Learning algorithms to generate employee assessment information, the EFPA guidelines on reliability and validity are no longer of any relevance. Now is the time to adopt more realistic and more intellectually honest procedures to evaluate the reliability and validity of any assessment; perhaps embracing the three broad requirements/principles outlined in this article. In reality, little of any importance is lost by discarding test-theory psychometrics, as the precision claimed by so many psychometricians using them is now known to be illusory. In real-world practice, few experienced assessment practitioners pretend their psychological assessments convey more than “ordered-class” information about an individual. When practitioners speak of reliability, it is that concerned with the direct observation of occasion-to-occasion variability, not abstract internal consistency estimates or factor-loading-based estimates. With game-based assessments consisting of hundreds and sometimes thousands of discrete estimation points, where outcome “indexes’, indications, orders, or classes are constructed using machine-learning and/or algorithmic expert-rules, evaluators need to see evidence that the outcome “indicators” from such complex and dynamic assessments do indeed relate to and/or predict that which they are claimed to relate/predict. Such evidence bases need to be constructed using methods appropriate to the specific claims made, avoiding obsolete abstract test theory and analysis methodologies which assume psychological attributes are being assessed as quantities. The EFPA guidelines served their purpose for the previous generation of self-report and ability questionnaire assessments. However, “next generation” assessments demand different kinds of evidence-bases suited to the assessment of attributes in what is essentially a non-quantitative science of psychology.

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
