# Peer review of "The EFPA Test-Review Model: When Good Intentions Meet a Methodological Thought Disorder"

_behavsci, 2017, doi:10.3390/bs8010005_

Round 1

Reviewer 1 Report

Specific

1. To make the paper have a high impact (and I think it could), I would suggest coming at "re-doing" sections 10 and 11 of the EFPA Test Review Model from the standpoint of a practitioner. In the US, this means they have had <= 1 course on measurement, <= 1 course on statistics, and maybe 1-2 courses on psychological assessment. 

For example, I come to p. 17 and read "When working with orders or types, existing or new methods directly suited to demonstrating repeatability of order or type are used." How do I examine this? I don't think the limited measures and arbitrary thresholds in section 10 of the EFPA TRM are necessarily the way to go, but they are easy for practitioners to get an idea of acceptable vs. unacceptable. How would I know if the "repeatability of order or type" from a scale is sufficient? Or even how would "repeatability of order or type" even be indexed?

2. p. 17-18 Validation.  The only thing discussed here is, for lack of better terms, concurrent/predictive validity. This is important, but what about content or construct types of validity evidence? How does this look coming from a perspective that the what we are measuring is not quantitive? 

As an aside, the sentence "No more or less than providing empirical evidence in support of any claim that is made regarding the results reported by an assessment" (p. 17) should be engraved on the door of every test publisher. The latest editions of Wechsler Intelligence Scales are great examples of how this is not being done and no one is holding them accountable.

3. Does it make sense that the "Next Generation of Assessments" should even fall under being reviewed by EFPA TRM? The characteristics described on pp. 11-13 are so different from what is done in psychology (at least in clinical/educational assessment--perhaps I-O too?) that I would think this would call for an entirely different approach to ``test'' evaluation, esp. given the proprietary nature of them. So, while I agree that the EFPA TRM cannot be used for such assessments, I don't think it was ever (or will ever?) designed to evaluate them.

4. The statements about legal challenges may be a bit overstated. I scanned through the latest version of Ziskin's text (6th edition, edited by Faust), and I couldn't find anything in there about the measurement debate (i.e., are the constructs even quantitative?) and I didn't see Michell listed as an "authority." This isn't to say that it won't be in the next edition or that the issues discussed in this paper are not informative for a legal context, just that I don't see the are-the-constructs-even-quantitative issue being part of many legal challenges in the immediate (Perhaps I am wrong, here; if so, the author should give some example legal cases where this has been an issue)

Minor

1. p. 14 "the very issue that Trendler [43, 44, 19]...Trendler [19] has"

I think the 19 should be 20

2. Reference [20].. "Conjoint Measurement undone" 

Is this accessible to the public somewhere? I tried to find it, but could not. What happens if it is not published?

Author Response

I've attached a pdf of my reply 

Reviewer 2 Report

The purpose of this study is to challenge the test standards and guidelines issued by the European Federation of Psychologists’ Associations (EFPA) in 2013 and provide some new framework for researchers. In general, this paper is well written.

Some of the issues pointed out by the author have been discussed in the literature, such as the measurement scale and the latent variable models as validation tools. In this paper the author goes further and challenges the foundation of the whole psychometric filed, challenging the concepts and assumptions of reliability and validity as well as the methods and procedures widely adopted by researchers. The author has made many interesting and good points. Personally I also found I am troubled by the existent validation and reliability estimation methods in some of my psychometric work. I agree with the author that we should redefine reliability and validity and discuss the issues that we have been facing for years. We should also provide some solutions and call for new approaches to meet the needs of newly appeared or appearing assessment tools. I would like to discuss a few issues brought up by the author.

1. Can the author make it clear what is new about your “scientific framework?

Lines 557-570: The author proposed his “scientific framework”, which basically includes phenomenon or indicators detection, causal explanatory theory construction, defining the construct under the study, and identifying the causes to explain the variations in the proposed measure of the construct. However, I don’t find anything new here. This procedure is not much different from what researchers have been practicing. For example, researchers go through the same procedure to develop their theory models via EFA.

2. Can the author clarify the measurement scale issue?

The author challenges the measurement scales used currently in psychometrics. For example, in Lines 671-674  the author pointed out, “…classical and modern test theory psychometrics was predicated on the assumption that all psychological attribute magnitudes vary as continuous or integer equal-interval entities;” This is not true. For instance, researchers in psychometrics and statistics have developed discrete latent variables, such as latent class analysis and latent profile analysis, which assumes that the underlying variables are categorical.

Furthermore, in Lines 572-587 the description of his procedure for the scientific framework still falls in the conventional measurement scales though the procedure is modified.

3. One minor suggestion: It may be better to rephrase the sentence, “To do requires a very technical/explicit definition of the construct…” (Lines 590-592)

4. Can the author clarify what he suggested in “R2: Evaluate Reliability” (Lines723-753)?

Would it be practical or sufficient if we only had one type of reliability estimation method, i.e., test-retest? If they didn’t have repeated measures design, researchers would never be able to provide reliability evidence. In addition, if we don’t use Pearson-based correlations, what specific methods or procedures would the author suggest? I think the suggestions from the author for reliability is fuzzy, not sufficient and the suggestion on eliminating all other types of reliability estimation methods is too extreme.

5. A general comment on quantitative latent variables:

Psychologists face a challenge—measuring something that cannot be measured directly, so that they have to construct latent variables and add meanings to these variables. I agree partly with the author on his view of latent variables. Sometimes, it is hard to quantify these variables and put them on equal interval or ratio scales, so we have to put them on ordinal or nominal scales. However, if researchers can collect enough evidence and make sense out of the quantitative latent variables, we should acknowledge this type of quantitative latent variables.

6. A general comment on IRT model:

The author provided several examples to demonstrate that Rasch IRT model is a failure and suggested to abandon it. Personally I am not a fan of Rasch model either, but I do not want to exclude this tool. It is still useful in some situations. This is similar to how doctors diagnose a patient’s problem. They will use different tools and collect a variety of evidence for their diagnosis.

When we conduct a scientific research project, we don’t want to only show one piece of evidence and make a conclusion. The machinery numbers we obtain from Rasch model may just give us nuisance like what the author and Wood have demonstrated, but they may provide useful information. We should use them together with other evidence or information and make a judgement about whether the numbers make sense or not. 

Author Response

I've attached a pdf of my reply
